# The clinicopathology and survival characteristics of patients with POLE proofreading mutations in endometrial carcinoma: A systematic review and meta-analysis

Alaa Salah Jumaah[1], Hawraa Sahib Al-Haddad[2], Katherine Ann McAllister[3]*, Akeel Abed Yasseen[1]

1 Department of Pathology and Forensic Medicine, Faculty of Medicine, University of Kufa, Kufa, Iraq, 2 Al-Furat Al-Awsat Hospital, Kufa, Najaf Governorate, Iraq, 3 School of Biomedical Sciences, Ulster University, Coleraine, Northern Ireland, United Kingdom

* k.mcallister@ulster.ac.uk

**Data Availability Statement:** All relevant data are within the manuscript and its Supporting Information files.

## Abstract

### Background

Endometrial carcinoma (EC) is classified into four distinct molecular subgroups. Patients with polymerase epsilon exonuclease domain mutated (POLE-EDM) tumors have the best prognosis of all. This meta-analysis consolidated the clinicopathology variations reported in the POLE-mutant subtype and survival parameters in patients with EC.

### Methods

The following internet data bases were searched: PubMed, Web of science, Embase and Scimage directory. Data was extracted from eligible studies including sample size, number of positive POLE-mutant cases, EDM sequencing information, clinicopathologic, and survival data. Meta-analysis and a random-effects model produced pooled estimates of POLE prognostic parameters using 95% confidence intervals (CI), hazard ratios (HR), and odds ratios (OR).

### Results

The meta-analysis included 11 cohort studies comprising 5508 EC patients (442 POLE EDM tumors). Patients with POLE mutant EC were associated with improved disease specific survival (HR = 0.408, 95% CI: 0.306 to 0.543) and progression-free survival (HR = 0.231, 95% CI: 0.117 to 0.456). POLE-mutated tumors were mostly endometrioid histology (84.480%; 95% CI: 77.237 to 90.548), although not significantly more than wild type tumors (OR = 1.386; p = 0.073). The POLE mutant tumors significantly present (p<0.001) at Federation of International of Gynecologists and Obstetricians (FIGO) lower stages I-II (OR = 2.955, p<0.001) and highest grade III (OR = 1.717, P = 0.003). The tumors are significantly

**Funding:** The author(s) received no specific funding for this work

**Competing interests:** The authors have declared that no competing interests exist.

associated with invasion less than half (<50%) of the myometrium (OR = 1.765, p = 0.001), but not deeply invasive EC (MI>50%, OR = 0.83, p = 0.34). POLE mutations significantly protected against lymph node metastases (OR = 0.202, p = 0.001), and have no clear association with lymph-vascular space invasion (OR = 0.967, 95% 0.713–1.310, p = 0.826). The tumors are predominantly of low ESMO risk stratification distribution (40.356%; 95% CI: 27.577 to 53.838).

## Conclusions

POLE mutations serve as an important biomarker of favorable prognosis in EC. The tumors are characteristically high grade, early stage, and remain localized in the endometrium with reduced likelihood of lymph node metastasis for improved survival prospects and the lowest risk classification. These findings have implications for medical management of EC.

## Introduction

Endometrial cancer (EC) is the most common gynecologic cancer in the USA, Japan and developing countries [1, 2]. Most cases (80%) are diagnosed at an early stage and have a 5-year overall survival rate of 95% [3]. However patients with advanced and recurrent disease have poor prognosis, even with surgery and adjuvant therapy [4]. Mortality occurs in over 20% of patients with EC and deathrates are increasing [5]. EC is stratified according to risk of recurrence to guide treatment plans for women. Clinicopathology risk assessments are based on tumor stage, grade, and histology subtype [3]. However there are reproducibility issues associated with traditional histopathology analysis of endometrial tumors among women [6, 7]. Use of molecular classification of EC offers an opportunity to improve risk assessments and treatment of women, especially over usage and underusage of adjuvant therapy [8].

EC was first classified into four distinct molecular subgroups by the Cancer Genome Atlas (TCGA) [9]. Later development of the Proactive Molecular Risk Classifier for Endometrial Cancer (ProMisE) used surrogate biomarkers to assign patients into the TCGA subtypes [10–12]. This simplified approach uses focused sequencing to detect POLE exonuclease domain mutations and immunohistochemistry identification of p53 and MMR proteins [13]. The four ProMise subtypes are: POLE exonuclease domain mutant (POLE EDM), MMR defective, non-specific molecular profile (p53 wild type), and p53 mutated tumors.

The POLE EDM mutant subtype has attracted attention with its favorable survival outcomes. There are two major hotspots for POLE proofreading mutations within the exonuclease domain: amino acid residue 286 (encoded by exon 9), and residue 411 (encoded by exon 13) [14, 15]. This causes the POLE EDM subtype and hypermutated tumors in EC patients [16]. Intriguingly, despite reports of poor clinicopathology the POLE ultra-mutated subtype has the best prognosis among patients with endometrial tumors; with a 5-year DSS rate of 98–100% [9, 17–19]. These tumors have unique intrinsic properties, are usually noted to be of endometrioid type and are enriched with infiltrated lymphocytes [18, 19].

Recently we reported the frequency of mutant POLE at 8.59% in EC using meta-analysis to pool 25 studies [20]. Our study found that POLE mutant tumors mainly presented at earlier stages I-II (89.51%) and at the highest grade III (51.53%) [20]. Another recent meta-analysis used six cohort studies confirmed the prognostic value of POLE exonuclease domain mutations (EDMs) for survival outcomes in 179 EC patients [21]. However, in relation to

clinicopathology, POLE EDMs were only found associated with early stage FIGO I group but not tumor grade, lymph-vascular space invasion (LVI), depth of myometrial invasion (MI), lymph node status and European Society for Medical Oncology (ESMO) risk groups [21]. Other studies have reported that POLE mutant EC tumors can be associated with higher grade [9, 11, 20] or not [10]. One study cohort of 544 EC tumors found no association between POLE mutation and progression-free survival [22]. These conflicting findings warrant further investigation.

This current meta-analysis was designed to clarify the survival analysis in POLE mutated EC in relation to clinicopathologic prognostic characteristics. We identified an expanded cohort of 11 studies to investigate patient survival in POLE mutant EC [8, 10–12, 22–28]— presently the largest number of patient cases. All the tumors were confirmed by EDM sequencing, according to the ProMisE protocols [11, 12, 18, 29]. The present study aimed to: (1) resolve the reported clinicopathology variations of POLE-mutant endometrial carcinoma and confirm (2) the prognostic benefit of the POLE (exonuclease domain mutant) subtype using survival analysis of the expanded cohort of studies.

## Methods

### Study protocol

This study was conducted according to the Preferred Reporting Item for Systematic Reviews and Meta-analyses (PRISMA) statement [30] (see S1 Checklist). The protocol methods for collection, data extraction and meta-analysis were developed. All stages of the review process were performed by two independent reviewers (ASJ & HSH). Any disagreement was solved by discussion, otherwise a third reviewer (AAA) was consulted.

### Search strategy

The internet literature review searched PubMed, web of science, and Embase. All oncology and pathology journals available in Scimago websites were searched. There was no cut-off date applied to the study. The following search terms were applied: "endometrial carcinoma", "endometrial cancer", "POLE EDM mutations", "EC", "POLE mutant,", "clinical characteristics" and "prognosis". All references in the included studies were screened for potentially suitable published articles. There were no language restrictions, however all eligible studies were published in English.

### The inclusion criteria

Two reviewers (ASJ & HSH) assessed the titles and abstracts. Full text papers were obtained for potentially eligible publications. The eligibility criteria were applied by two independent reviewers (ASJ & HSH). Any disagreement was solved by consensus otherwise consulting with senior reviewer (AAA). Inclusion criteria are:

a. POLE mutation was tested by gene sequencing in the articles.

b. Adequate clinicopathologic data was available as: Federation of International of Gynecologists and Obstetricians (FIGO) pathological staging and grading, lymphovascular invasion (LVI), histologic variants, extent of myometrial invasions (MI), lymph node metastasis, overall survival (OS), disease specific survival (DSS) and progression free survival (PFS).

c. There were sufficient data to extract the parameters: hazard ratio (HR) and its standard error (SR), and to calculate odds ratios (OR)

## The exclusion criteria

a. There was not enough data for calculation.

b. Patients were not confirmed POLE mutant by EDM sequencing.

c. Duplication of the publication.

d. Single case reports, commentaries, editorials, letters to the editors, review articles, and unrelated articles.

## Data extraction and measured outcomes

Two authors (ASJ & HSH) extracted the data independently. The following parameters were extracted from the studies: first author name, publication year, total number of EC, number of POLE mutant EC, study country of origin, histological type, FIGO stage and grade, LVI, extent of MI, lymph node involvement, European Society for Medical Oncology (ESMO) risk stratification, HR and its 95% confidence interval (95% CI). The ESMO guidelines categorizes the risk of recurrence into (1) low, (2) intermediate, and (3) high-risk groups by tumor stage, grade, and histology subtype [31]. For example, about 75% of patients present with stage I disease and can be subdivided into three risk categories with regard to disease relapse and survival:

1. low risk: stage Ia/Ib, grade 1 or 2, endometrioid histology

2. intermediate risk: stage Ic, grade 1 or 2, endometrioid histology; stage Ia/Ib, grade 3, endometrioid histology

3. high risk: stage Ic, grade 3, endometrioid histology; stage Ia or Ib or Ic, serous, clear cell, small cell or undifferentiated histology."

Any disagreements were solved by consensus under the supervision of senior author (AAA).

## Statistical analysis

The data was analyzed by MedCalc Statistical Software version 15.8. Statistical heterogeneity of the 11 included studies was assessed using the $I^2$ test. The following prognostic parameters were estimated with HR: OS, DSS, and PFS. Clinicopathologic variables were compared between POLE-mutant and wild type EC patients using ORs. All statistics were reported with their 95% CI. $I^2$ inconsistency test and the chi-squared-based Cochran Q statistic test were used to investigate heterogeneity [32]. The random effect model was used when $I^2 > 50\%$ (indicating significant heterogeneity) and when $I^2 < 50\%$ the fixed effect model was used. Subgroup analysis was used to investigate the heterogeneity source. The funnel plot test was used to assess publication bias [33].

## Study quality assessment

The Quality Assessment Tool for Diagnostic Accuracy Studies-2 (QUADAS-2) was used for bias assessment [34]. This tool consisted of several signaling questions in relation to the study design in respect of four domains including "patient selection", "index test", "reference standard", "flow and timing" and, another three domains for study applicability that included "patient selection", "index test", and "reference standard". The risk of bias and applicability concerns were investigated and checked by signaling questions and labeled as "yes", "no" or "unclear". The result labelled the risk of bias as "high", "low" or "unclear".

## Sensitivity and subgroup analysis

Sensitivity analysis was conducted by omitting each study one-by-one to see its contribution on the pooled meta-analysis results. Subgroup analysis was performed according to geographical area of Canada, USA and Europe to discover sources of heterogeneity.

## Ethics

This meta-analysis was approved by the Institutional Review Board of the University of Kufa (IRB approval No. UK-2019-0534). Formal written informed consent was not required with a waiver issued by the Institutional Review Board of the University of Kufa. All the authors are responsible for any false statements or failure to follow the ethical guidelines. The authors are accountable for all aspects of the work in ensuring that questions related to the accuracy or integrity of any part of the work are appropriately investigated and resolved. All procedures performed in studies involving human participants were in accordance with the ethical standards of the institutional and/or national research committee(s) and with the Helsinki Declaration (as revised in 2013).

# Results

## Study characteristics and quality assessment

During the initial search, 378 studies were identified, of which 283 were excluded. The remaining 95 studies were full text screened for eligibility according to the protocol criteria. A total of 11 studies met the inclusion criteria for meta-analysis, as shown in Fig 1.

After screening, the quality of the included studies was assessed according to QUADAS-2. Most studies displayed a low risk of bias and applicability concern (S1 and S2 Figs). The characteristics of the included studies are presented in Table 1. There were a total number of 5503 patients with EC across the studies. The proportion of patients with POLE mutant tumors ranged from 12 to 49, with a total of 442 patients across the studies. The included articles were published between 2015 and 2019. All the studies were performed in Canada, the USA, or European countries. The POLE mutations were identified by POLE exonuclease domain (EDM) sequencing.

## Survival meta-analysis in POLE mutant EC

OS was assessed using data extracted from ten studies, eight for DSS, and six for PFS, in patients with POLE mutant EC. The meta-analyses of hazard ratios for survival outcomes OS, DSS, and PFS are shown in Fig 2A–2C). There was no significant heterogeneity ($I^2$ = 0.00%) for OS, DSS and PFS across the studies, and a fixed effect model was selected for analysis. Fig 2D shows the meta-analysis for the proportion of POLE mutated EC in the studies (Fig 2D).

Patients with POLE mutant tumors were associated with improved survival, according to the parameters DSS (HR = 0.408, 95% CI:0.306 to 0.543) and PFS (HR = 0.231, 95% CI: 0.117 to 0.456) (Table 2). However the OS outcome in POLE mutant EC patients was unclear (HR = 0.772, 95% CI:0.574 to 1.039). While the HR was favorable for OS, the confidence interval had a wide range and crosses one.

## POLE mutated EC frequency and subgroup analysis

The pooled proportion of POLE mutated EC was 8.526% (95% CI: 7.143 to 10.018) (Table 3, Fig 2D). The random effect model was applied for final meta-analysis since there was significant heterogeneity between studies ($I^2$ = 69.33%, 95% CI: 45.67 to 82.69, P = 0.001). Publication bias was visualized using the funnel plot (Fig 3). A subgroup analysis of POLE mutant EC according to country of origin was performed to explore the cause of heterogeneity. The

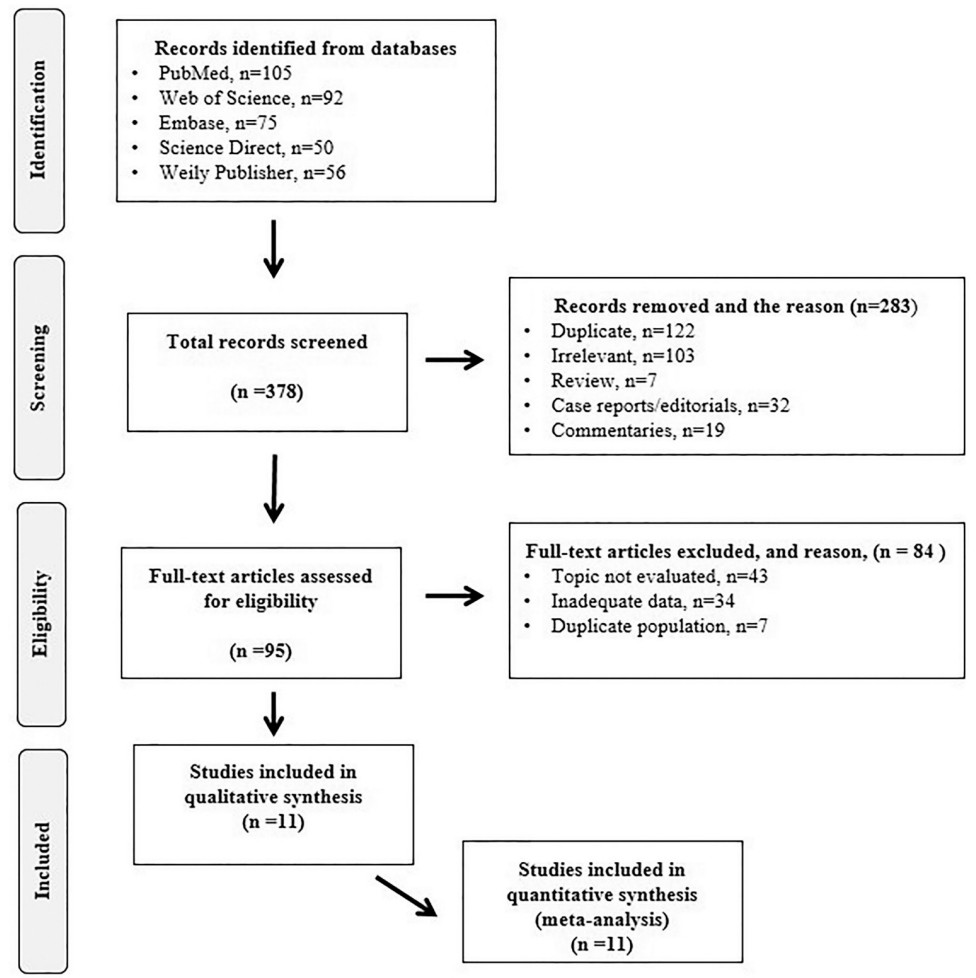

**Fig 1. PRISMA flowchart showing the literature search and selection process.**

studies were divided into 2 groups according to the geographical area. There was slightly higher heterogeneity in European studies $I^2$ = 69.570% (95% CI: 22.140 to 88.110) compared to studies performed in the USA and Canada $I^2$ = 64.320% (95% CI: 19.510 to 84.190) (Fig 4).

## Sensitivity analysis

Sensitivity analysis was performed for the POLE mutant EC studies and those specific cases involved in estimating OS, DSS and PFS (by removing each of the studies in turn from the final pooled analysis). This assesses the influence of the removed dataset on the pooled HR. The results were robust and not significantly affected by exclusion of any studies (refer to the S1–S4 Tables).

## Clinicopathologic parameters in POLE mutant EC

Clinicopathology data was extracted from eligible studies of POLE mutant EC for meta-analysis (S3–S8 Figs). The pooled proportions of FIGO stage, FIGO grade, extent of MI, LVI, lymph node (LN) involvement and ESMO risk stratification are reported in Table 3 for patients with POLE mutant EC. The pooled odds ratios were also calculated for POLE-mutant versus wild type POLE according to each clinicopathologic variable (Table 4).

## FIGO stage and grade: Pooled proportion and odds ratio

The eligible studies used to pool proportions and odds ratios for FIGO stage and grade are shown in S3A–S3D and S4A–S4D Figs. The pooled stage I-II in POLE mutant EC is 92.026% (95% CI: 86.143 to 96.392) while that of stage III-IV is 4.970% (95% CI: 2.795 to 7.727), as documented in Table 3. The pooled odds ratio of stage I-II in POLE mutant EC to wild type POLE EC is 2.955% (95% CI: 1.937 to 4.507) and that for stage III-IV is 0.187% (95% CI: 0.107 to 0.325), as also shown in Table 4.

Meanwhile the pooled proportion of grade I-II in POLE mutant EC is 52.724% (95% CI: 38.735 to 66.499) and that of grade III is 43.439 (95% CI: 28.491 to 59.025). The odds ratio for POLE mutant EC to wild type POLE EC for grade I-II is 0.514 (95% CI: 0.397 to 0.664) and that for grade III is 1.717 (95% CI: 1.209 to 2.439). (Refer to Tables 3 and 4). These results conclude that POLE mutant tumors significantly present ($p < 0.001$, $P = 0.003$) at both the lower stages I-II and highest grade III when compared to wild type tumors.

## Myometrial invasion (MI)

The eligible studies used to pool proportions and odds ratios for MI are shown in S5A–S5D Fig. The pooled proportion of MI<50% is POLE mutant EC is 49.157% (95% CI: 41.238 to 57.096) and that of MI>50% is 38.398% (95% CI: 28.588 to 48.710), as shown in Table 3. Meanwhile the odds ratio of MI in POLE mutant EC to MI in wild type POLE EC in Table 4 is: MI<50% is 1.765 (95% CI: 1.280 to 2.435) and for MI>50% is 0.826 (95% CI: 0.559 to 1.221). POLE mutations are only significantly associated with invasion less than one half (<50%) of the myometrium ($p = 0.001$) relative to wild type tumors (Table 4). However, this tendency is lost during advanced disease. There is no clear association of POLE mutations and deep MI >50% ($p = 0.34$).

## Lymphovascular invasion (LVI)

The pooled proportion of LVI in POLE mutant EC is 22.324% (95% CI: 7.716 to 41.771). On the other hand the odds ratio of LVI in POLE mutant EC to wild type POLE EC is 0.967 (95%

**Table 1. Study characteristics.**

| Study Author & Publication Year | Study Type | EC Cohort Size | POLE-Mutant Numbers | Study Country | Sequencing Method | Location of Exonuclease Mutations | Outcome |
|---|---|---|---|---|---|---|---|
| Kommoss et al, 2018 | Cohort | 452 | 47 | Germany | Targeted next generation sequencing | Exons 9–14 | OS, DSS |
| | | | | | | | PFS |
| Billingsley et al, 2015 | Cohort | 535 | 30 | USA | PCR, sanger sequencing | NR | OS, PFS |
| Talhouk et al, 2017 | Cohort | 319 | 30 | Canada | Sequencing | NR | OS, DSS, PFS |
| Stelloo et al, 2016 | Cohort | 834 | 49 | Netherlands | Sanger sequencing | Exons 9 and 13 | OS |
| Talhouk et al, 2015 | Cohort | 143 | 12 | Canada | Sequencing | Exon 12 | OS, DSS |
| Church et al, 2015 | Cohort | 788 | 48 | Europe | Sequencing | Exon 9 and 13 | OS, DSS |
| Proctor et al, 2017 | Cohort | 90 | 14 | Canada | Sequencing | Exon 12 | OS, DSS |
| | | | | | | | PFS |
| Talhouk et al, 2018 | Cohort | 460 | 42 | Canada | Sequencing | NR | OS, DSS, PFS |
| Imboden et al,2019 | Cohort | 599 | 38 | Sweden | Sequencing | Exons 9–14 | DSS, PFS |
| Karnezis et al, 2017 | Cohort | 460 | 42 | Canada | Sequencing | Exons 9–14 | OS, DSS, PFS |
| Bosse et al, 2018 | Cohort | 376 | 48 | Europe and USA | Sanger & next-generation | Exons 9–14 | OS, PFS |

EC, endometrial carcinoma; OS, overall survival; DSS, disease specific survival; PFS, progression free survival.

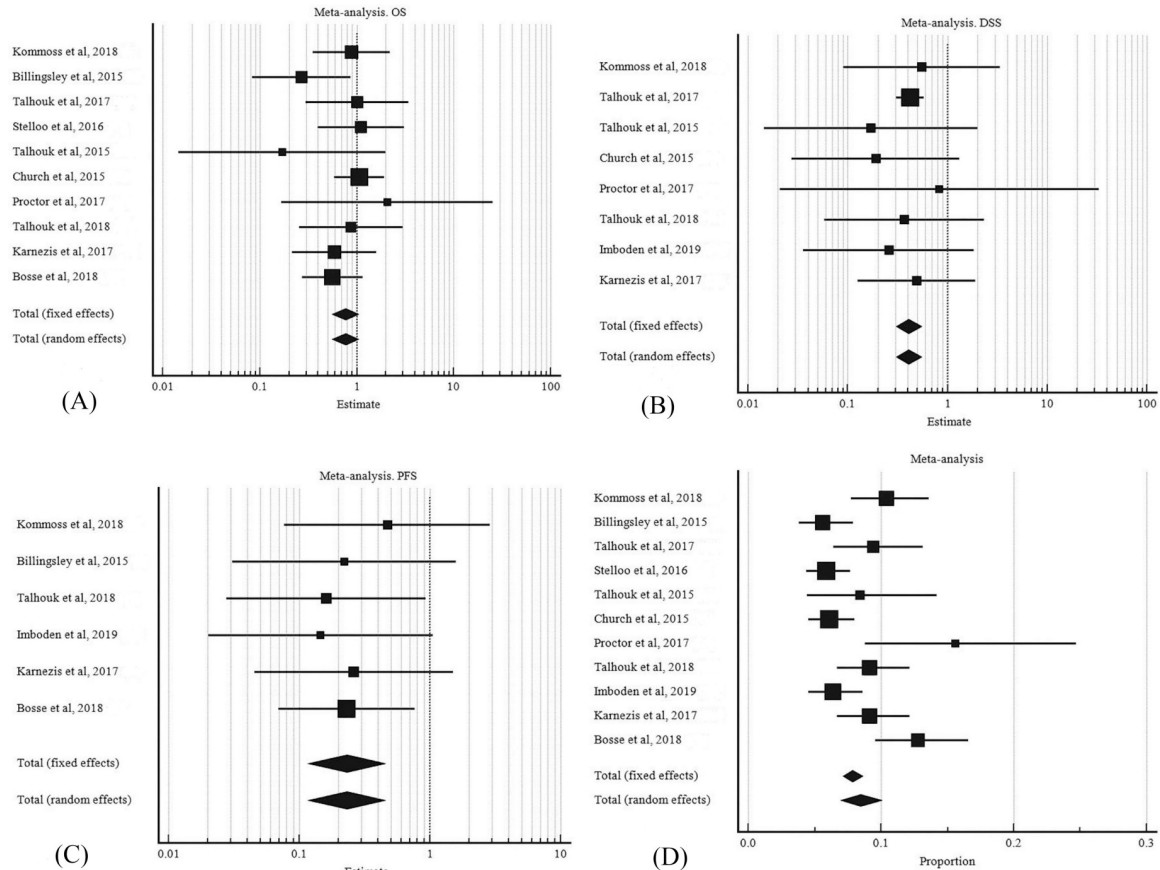

**Fig 2. POLE mutations and survival analysis meta-analysis.** A, overall survival. B, disease specific survival. C, progression free survival. D, proportion of POLE mutation in endometrial carcinoma.

CI: 0.713 to 1.310), with a non-significant p-value of 0.826. This shows that POLE mutation does not influence the tendency of EC to invade lymphovacular spaces. The eligible studies used to pool proportions and odds ratios for LVI in POLE mutant EC are shown in S6A, S6B Fig, and the pathology characteristics and odds ratios are shown in Tables 3 and 4.

## Lymph nodes metastases

S7A–S7D Fig shows the studies used to estimate pooled proportions and odds ratios, and the resultant data in Tables 3 and 4. The pooled proportion of positive lymph nodes in POLE mutant EC 1.282% (95% CI: 00.243 to 3.838) while for negative is 74.330% (95% CI: 61.789 to 85.106). The pooled odds ratio of positive lymph nodes in POLE mutant EC to wild type POLE EC is 0.202 (95% CI: 0.078 to 0.519) and that for negative nodes is 2.070 (95% CI: 1.499 to 2.858). These significant findings (p≤0.001) demonstrate the protective effect of POLE mutation against lymph node metastases.

## Histologic variants

The pooled proportion of endometrioid type in POLE mutant EC is 84.480% (95% CI: 77.237 to 90.548) while that for non-endometrioid type is 12.437% (95% CI: .473 to 18.447). The pooled odds ratio of endometrioid histology in POLE mutant EC compared to wild type tumors is 1.386 (p = 0.073) and that for the non-endometrioid variant is 0.582 (p = 0.007). The

**Table 2. Survival analysis in POLE mutated EC.**

| Study | OS | DSS | PFS | Survival analysis test | Method |
|---|---|---|---|---|---|
| | estimated HR (95% CI) | estimated HR (95% CI) | estimated HR (95% CI) | | |
| Kommoss et al, 2018 | 0.878 (0.351 to 2.200) | 0.550 (0.0900 to 3.361) | 0.470 (0.0769 to 2.874) | Multivariable survival analysis | Kaplan–Meier & cox proportional hazard models |
| Billingsley et al, 2015 | 0.270 (0.0838 to 0.870) | | 0.220 (0.0306 to 1.581) | Multivariable Analysis | Kaplan-Meier estimates |
| Talhouk et al, 2017 | 1.010 (0.298 to 3.425) | 0.420 (0.306 to 0.576) | | Multivariable survival analyses | Cox proportional-hazards model |
| Stelloo et al, 2016 | 0.170 (0.0146 to 1.983) | | | Multivariable analysis | Kaplan-Meier method & log-rank test starting |
| Talhouk et al, 2015 | 0.170 (0.0146 to 1.983) | 0.170 (0.0145 to 1.997) | | Multivariable analyses | Kaplan–Meier survival analyses & log-rank statistics |
| Church et al, 2015 | 1.060 (0.588 to 1.912) | 0.190 (0.0274 to 1.316) | | Multivariable analysis | Kaplan-Meier method & log-rank test comparisons |
| Proctor et al, 2017 | 2.060 (0.166 to 25.495) | 0.830 (0.0210 to 32.803) | | Univariable survival analyses | Kaplan-Meier curve |
| Talhouk et al, 2018 | 0.870 (0.254 to 2.985) | 0.370 (0.0583 to 2.347) | 0.160 (0.0276 to 0.926) | multivariable Cox proportional hazard models | Kaplan Meier method |
| Imboden et al, 2019 | | 0.258 (0.0359 to 1.855) | 0.145 (0.0201 to 1.047) | | Kaplan-Meier curves |
| Karnezis et al, 2017 | 0.590 (0.217 to 1.607) | 0.490 (0.126 to 1.904) | 0.260 (0.0453 to 1.494) | Univariable survival analysis | Kaplan-Meier survival analyses |
| Bosse et al, 2019 | 0.560 (0.271 to 1.156) | | 0.230 (0.0693 to 0.763) | Multivariable analyses | Kaplan-Meier survival analyses |
| Pooled HR (95% CI) | 0.772 (0.574 to 1.039) | 0.408 (0.306 to 0.543) | 0.231 (0.117 to 0.456) | | |
| $I^2$ (95% CI) | 0.00% (0.00 to 50.92) | 0.00% (0.00 to 0.00) | 0.00% (0.00 to 0.00) | | |

EC, endometrial carcinoma; OS, overall survival; DSS, disease specific survival; PFS, progression free survival.

findings suggest that POLE mutant EC have mainly endometrioid histology, but not significantly so when compared to wild type tumors (Refer to Tables 3 and 4 and S8A–S8D Fig).

### ESMO risk stratification

Table 3 shows that the pooled proportion for low ESMO POLE mutant EC is 40.356% (95% CI: 27.577 to 53.838), intermediate ESMO POLE mutant EC is 21.737% (95% CI: 8.885 to 38.306), high ESMO POLE mutant EC is 26.401% (95% CI: 13.117 to 42.356). These findings show that most cases of POLE mutant EC present as low risk ESMO.

### Discussion

This updated meta-analysis reports clinicopathology characteristics and survival outcomes in patients with the POLE EDM subtype of endometrial tumor. The findings indicate that the prognosis and future well-being of women with POLE-mutant EC holds great promise. Despite the alarming presentation with FIGO grade III tumors, patients with POLE mutant EC have improved PFS, DSS, and are mainly classified in the lowest ESMO risk group. Endometrial tumors with the POLE EDM subtype tumors are typically low FIGO stage. The clinicopathology meta-analysis findings also suggest that POLE mutant tumors remain localized at the endometrium, without deep progression into the muscular myometrial layer, with reduced

**Table 3. The association between POLE mutated EC and clinicopathologic characteristics.**

| Clinicopathology characteristics in POLE mutant EC | Study Numbers | Pooled % portion (95% CI) | I² (95% CI) | P-value | Model |
|---|---|---|---|---|---|
| Overall POLE mutation in EC | 11 | 8.545 (7.212 to 9.979) | 69.33% (45.67 to 82.69) | 0.001 | Random effect |
| Stage I-II | 8 | 92.026 (86.143 to 96.392) | 62.94% (20.31 to 82.77) | 0.008 | Random effect |
| Stage III-IV | 8 | 4.970 (2.795 to 7.727) | 0.00% (0.00 to 67.72) | 0.433 | Fixed effect |
| Grade I-II | 8 | 52.724 (38.735 to 66.499) | 84.20% (70.55 to 91.52) | < 0.001 | Random effect |
| Grade III | 8 | 43.439 (28.491 to 59.025) | 87.23% (77.03 to 92.91) | < 0.001 | Random effect |
| Lymphovascular invasion | 8 | 22.324 (7.716 to 41.771) | 92.80% (88.13 to 95.63) | < 0.001 | Random effect |
| Myometrial invasion less than 50% | 7 | 49.157 (41.238 to 57.096) | 47.37% (0.00 to 77.79) | 0.076 | Random effect |
| Myometrial invasion more than 50% | 7 | 38.398 (28.588 to 48.710) | 69.25% (32.32 to 86.03) | 0.003 | Random effect |
| Lymph nodes positive | 6 | 1.282 (0.243 to 3.838) | 0.00% (0.00 to 69.27) | 0.547 | Fixed effect |
| Lymph nodes negative | 6 | 74.330 (61.789 to 85.106) | 74.60% (42.31 to 88.82) | 0.001 | Random effect |
| Endometrioid | 7 | 84.480 (77.237 to 90.548) | 55.98% (0.00 to 81.08) | 0.034 | Random effect |
| Non-endometrioid | 7 | 12.437 (7.473 to 18.447) | 46.28% (0.00 to 77.36) | 0.083 | Random effect |
| Low ESMO risk | 6 | 40.356 (27.577 to 53.838) | 75.44% (44.57 to 89.12) | 0.001 | Random effect |
| Intermediate ESMO risk | 6 | 21.737 (8.885 to 38.306) | 86.54% (72.91 to 93.31) | < 0.001 | Random effect |
| High ESMO risk | 6 | 26.401 (13.117 to 42.356) | 84.31% (67.49 to 92.43) | < 0.001 | Random effect |

The denominator for overall POLE mutation includes all patients with EC; whereas for subsequent clinicopathological parameters, the denominator is limited to patients with POLE mutated EC. EC, endometrial carcinoma; I², The statistic that indicates the percentage of variance in a meta-analysis that is attributable to study heterogeneity; ESMO, The European Society for Medical Oncology. Myometrial invasion is expressed as invasion of either < 50%> of the myometrium (50%MI) according to the FIGO staging system.

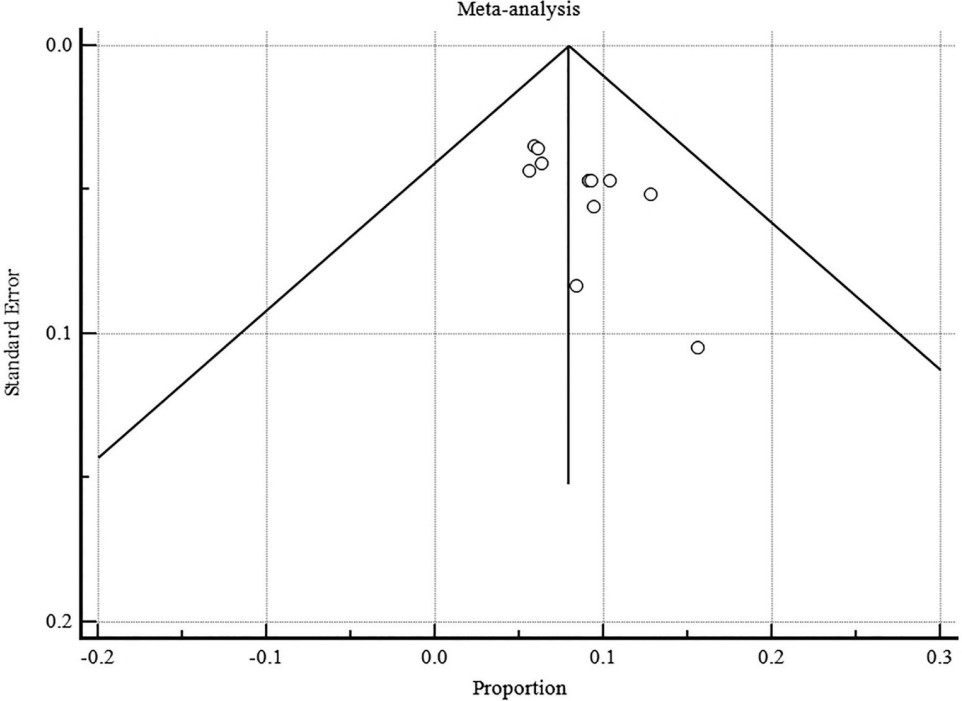

**Fig 3. Funnel plot for publication bias.**

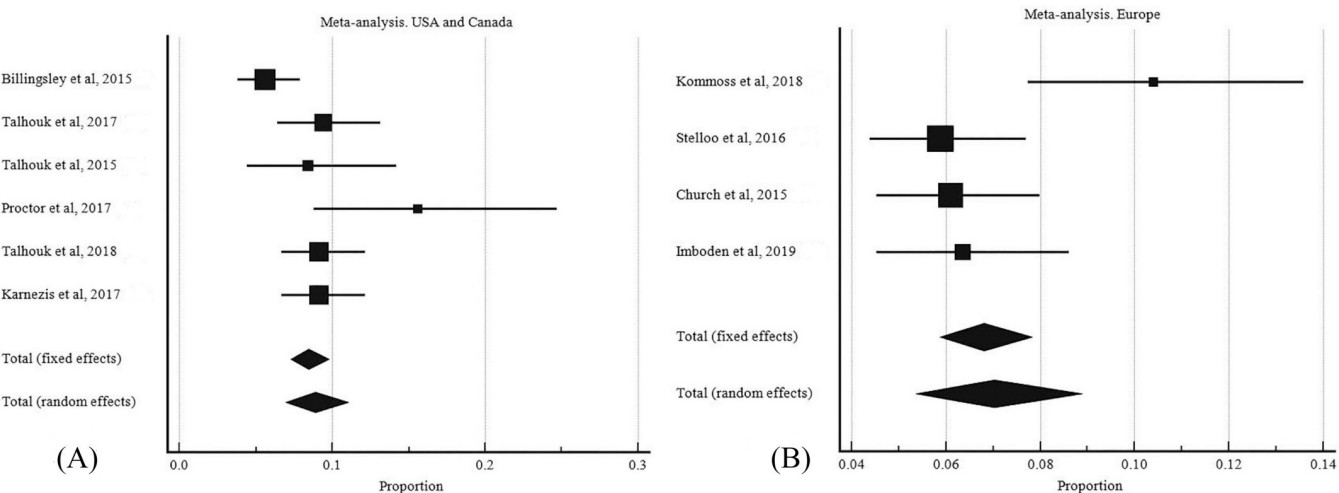

**Fig 4. Subgroup analysis for POLE mutant EC.** A, United States and Canada. B, European studies.

likelihood of metastasis since the lymph nodes are mostly negative. In endometrial carcinoma, MI is a well-known predictor of recurrence, and an important factor in the decision-making process for adjuvant treatment [35]. Patients with more than 50% MI on gross visual intraoperative estimation are at high risk for extrauterine metastases, including pelvic and para-aortic lymph node metastases [36]. The current meta-analyses shows that POLE mutated EC tumor invasion is significantly curtailed to within 50% of the myometrium, there is no clear association with POLE mutations with invasion beyond 50% of the myometrium, and likelihood of positive lymph node metastases is reduced when compared to wild type POLE tumors (OR = 0.202, p = 0.001).

In principle, the loss of functional polymerase epsilon compromises the fidelity of DNA replication, causing an ultra-mutated high grade tumor phenotype in the immediate locality of the endometrium. Fortunately in this subtype other protective intrinsic tumor characteristics curtail the development and spread of cancer throughout the body. A recent study showed

**Table 4. Pooled odds ratio of clinicopathologic variables in POLE-mutant EC VS wild type tumors.**

| Clinicopathology: POLE-mutant VS Wild Type | Pooled OR (95% CI) | P-value | Study Number | I$^2$ (95% CI) | P-value for I$^2$ | Model |
|---|---|---|---|---|---|---|
| Stage I-II EC | 2.955 (1.937 to 4.507) | <0.001 | 8 | 20.59% (0.00 to 64.02) | 0.272 | Fixed effect |
| Stage III-IV EC | 0.187 (0.107 to 0.325) | <0.001 | 8 | 0.00% (0.00 to 43.25) | 0.804 | Fixed effect |
| Grade I-II EC | 0.514 (0.397 to 0.664) | <0.001 | 8 | 17.72% (0.00 to 60.59) | 0.289 | Fixed effect |
| Grade III EC | 1.717 (1.209 to 2.439) | 0.003 | 8 | 43.53% (0.00 to 75.01) | 0.088 | Random effect |
| LVI | 0.967 (0.713 to 1.310) | 0.826 | 8 | 4.05% (0.00 to 69.22) | 0.398 | Fixed effect |
| MI less than 50% | 1.765 (1.280 to 2.435) | 0.001 | 7 | 39.58% (0.00 to 74.59) | 0.001 | Random effect |
| MI more than 50% | 0.826 (0.559 to 1.221) | 0.337 | 7 | 55.49% (0.00 to 80.89) | 0.036 | Random effect |
| Endometrioid histology | 1.386 (0.970 to 1.979) | 0.073 | 7 | 34.13% (0.00 to 72.13) | 0.167 | Fixed effect |
| Non-endometrioid histology | 0.582 (0.392 to 0.863) | 0.007 | 7 | 0.00% (0.00 to 60.97) | 0.620 | Fixed effect |
| Lymph nodes positive | 0.202 (0.078 to 0.519) | 0.001 | 6 | 0.00% (0.00 to 30.58) | 0.879 | Fixed effect |
| Lymph nodes negative | 2.070 (1.499 to 2.858) | <0.001 | 6 | 0.00% (0.00 to 0.00) | 0.990 | Fixed effect |

EC, endometrial carcinoma; OR, odds ratio; LVI, lymphovascular invasion; MI, extent of myometrial invasion; I$^2$, The percentage of variance in a meta-analysis that is attributable to study heterogeneity.

that POLE mutations improve the prognosis of EC via regulation of cellular glucose metabolism through AMF/AMFR signal transduction [35]. The study also found enrichment of the T cell receptor signaling pathway and immune response mediators in the tumors [35]. POLE-mutant ECs are well characterized by CD4+ and CD8+ lymphocytic infiltrates, a gene signature of T cell infiltration, and marked upregulation of cytotoxic T cell effector markers [36, 37].

Metastasis is a complex systemic disease that develops from interactions between tumor cells and their local and distant microenvironments. Local and systemic immune-related changes also play critical roles in limiting or enabling the development of metastatic disease [38]. Indeed POLE mutated EC is characterized by high immune infiltrates that may be further stimulated by therapy [39] along with PD-1 and PD-L1 expression [40]. These immune cells may counteract the survival risk caused by high grade ultra-mutated POLE tumors that are immunogenic [36]. This is clearly evidenced in the updated meta-analysis with the improved survival outcomes (DSS and PFS) and clinicopathologic findings (low FIGO stage and ESMO risk groups, lack of positive lymph node involvement, with no clear association with LVSI or deep MI). Based on the findings of our study POLE-mutant status should be clarified at diagnosis so that less intensive adjuvant therapy is administered. Currently the risk classification of EC determines the treatment plan in the patient. For example, in the ESMO guidelines, low risk FIGO stage I EC cases do not require adjuvant therapy, while adjuvant pelvic radiotherapy is advisable in intermediate risk FIGO stage I EC for a significant reduction in pelvic/vaginal relapse. On other hand, pelvic radiotherapy is recommended in high risk FIGO stage I endometrial carcinoma for more effective loco-regional control [41]. The recent 2020 ESGO/ESTRO/ESP risk stratification has now integrated molecular classification and treatment algorithms [29] and will provide a new benchmark for care of patients with EC. One recommendation is that pathogenicity of identified POLE variants should be reported [42].

The result of the current meta-analysis were also compared to previous studies, including our prior meta-analysis [20] and that of He and colleagues [21]. Previously the vast majority of studies of POLE-mutant EC reported mainly FIGO stage I, grade 3, and endometrioid histology in most patients. The current meta-analysis found that POLE mutant tumors had mainly endometrioid histology, consistent with other studies [9, 20, 35, 43]. However odds ratio analysis found no significant difference in the likelihood of endometrioid histology in comparison to POLE wild type tumors. Prior meta-analysis [21] has also reported no significant differences in histology type in POLE-mutant EC (endometrioid versus non endometroid, P = .09). POLE mutant tumors significantly presented at high grade and low stage, as reported elsewhere [7, 9, 10, 15, 20, 24, 44]. Our finding that POLE EDM status did not significantly modify the tendency to invade lymphovascular spaces matches the analysis by He and colleagues [21]. This meta-analysis also determined favorable improved PFS and DSS in POLE EDM tumors, confirming the results of many prior studies [8, 10–12, 15, 24, 25, 27, 28, 37]. While the HR was also favorable for OS in the current investigation (HR = 0.772) and consistent with previous studies including prior meta-analysis [20, 21]—the confidence interval had a wide range and crosses one (95% CI:0.574 to 1.039). Therefore the influence of POLE EC mutations on patient OS is still uncertain, and larger sample sizes are required to determine effects.

Undoubtably the integration of molecular classification of EC with clinicopathology risk assessments is the future for risk prediction and treatment plans of patients. The PORTEC-4a trial is currently ongoing and will confirm if omitting treatment in cases of favorable molecular profiles such as POLE-EDM subtype is a safe and cost-effective approach [45]. In the meantime, meta-analysis also offers a valuable tool to synthesis evidence regarding tumor molecular subtypes such as POLE EDM across existing studies.

## Strengths and limitations

There are several factors that strengthen the results of the new meta-analysis. Firstly, a higher number of studies [12] was used for study estimates, overcoming the problems of small sample size and false negative results. Meta-analysis increases statistical power to determine even small and clinically significant effects by combining data from numerous studies. The precision of a given study's findings largely depend on the number of subjects. Therefore combining data from several studies will provide a more precise estimate of the effect under investigation than using just a single study. Secondly, this meta-analysis used studies from different populations (Europe, Canada, and the United States)—this increased the generalizability of the results to represents wider populations. Overall this study has expanded areas that lacked adequate evidence and the robustness of POLE-mutant EC survival and clinicopathologic characteristics. This study settles controversies between studies with conflicting findings. This provides a firm basis for investigating new research questions in this tumor subset.

There are still limitations in the meta-analysis study. The first limitation is that only 6 studies had adequate data to estimate pooled PFS. The second limitation was heterogeneity in calculated pooled proportions of POLE mutation in EC. Summarizing results and information from different studies by meta-analysis will generate heterogeneity, as different standardization approaches are used between studies to estimate the results. We tried to solve this problem by performing subgroup analysis according to geographic distribution of the studies and a random effect model. Another issue is publication bias and subjectivity in picking up the relevant studies. This is overcome by performing a meticulous search to select studies for meta-analysis and assess visually using the funnel plot test. This subjectivity was addressed by use of two study selection authors and any disagreement were solved by discussion between them or consensus with a third senior author.

## Conclusion

POLE mutation serves as an important biomarker of favorable prognosis. The tumors are typically high grade, early stage, and remain localized in the endometrium with reduced likelihood of lymph node metastasis for improved survival prospects and the lowest risk classification. The clarification of POLE-mutant status in patients with EC can help guide individualized medical treatment and prevent unnecessary use of therapy. Molecular classification should be considered in all ECs and be incorporated into management decisions.

## Supporting information

**S1 Checklist. PRISMA 2009 checklist.**
(PDF)

**S2 Checklist.**
(DOCX)

**S1 Fig. QUADAS-2 results for studies assessment.**
(TIF)

**S2 Fig. Risk of bias and applicability concern (QUADAS-2 studies).**
(PNG)

**S3 Fig. FIGO stage in POLE mutant EC. A**, pooled proportion of stage I-II. **B**, pooled proportion of stage III-IV. **C**, odds ratio of stage I-II POLE mutant EC to stage I-II wild type POLE EC. **D**, odds ratio of stage III-IV POLE mutant EC to stage III-I VS wild type POLE EC.
(DOCX)

**S4 Fig. FIGO grade in POLE mutant EC. A**, pooled proportion of grade I-II. **B**, pooled proportion of grade III. **C**, odds ratio of grade I-II POLE mutant EC to grade I-II wild type POLE EC. **D**, odds ratio of grade III POLE mutant EC to grade III wild type POLE EC.
(DOCX)

**S5 Fig. Myometrial invasion extent in POLE mutant EC. A**, pooled proportion MI<50%. **B**, pooled proportion of MI>50%. **C**, odds ratio of MI<50%. POLE mutant EC to MI<50% wild type POLE EC. **D**, odds ratio of MI>50% POLE mutant EC to MI>50% wild type POLE EC.
(DOCX)

**S6 Fig. LVI in POLE mutant EC. A**, pooled proportion LVI. **B**, odds ratio of LVI POLE mutant EC to LVI wild type POLE EC.
(DOCX)

**S7 Fig. LN involvements in POLE mutant EC. A**, pooled proportion LN negative **B**, pooled proportion of LN positive. **C**, odds ratio of LN negative POLE mutant EC to LN negative wild type POLE EC. **D**, odds ratio of LN positive POLE mutant EC to LN positive wild type POLE EC.
(DOCX)

**S8 Fig. Histology of POLE mutant EC. A**, pooled proportion of endometroid type POLE mutant EC **B**, pooled proportion of non-endometroid type POLE mutant EC. **C**, odds ratio of endometrioid POLE mutant EC to endometrioid wild type POLE EC. **D**, odds ratio of non-endometrioid POLE mutant EC to non-endometrioid wild type POLE EC.
(DOCX)

**S1 Table. Sensitivity analysis of POLE mutant EC.**
(DOCX)

**S2 Table. Sensitivity analysis for cases involved in the analysis of overall survival.**
(DOCX)

**S3 Table. Sensitivity analysis of disease specific survival studies.**
(DOCX)

**S4 Table. Sensitivity analysis of studies involved in the analysis of progression free survival.**
(DOCX)

**S1 File. Excel.**
(XLSX)

## Author Contributions

**Conceptualization:** Alaa Salah Jumaah, Hawraa Sahib Al-Haddad, Katherine Ann McAllister, Akeel Abed Yasseen.

**Data curation:** Alaa Salah Jumaah, Hawraa Sahib Al-Haddad.

**Formal analysis:** Alaa Salah Jumaah, Hawraa Sahib Al-Haddad, Katherine Ann McAllister, Akeel Abed Yasseen.

**Investigation:** Alaa Salah Jumaah, Hawraa Sahib Al-Haddad, Akeel Abed Yasseen.

**Methodology:** Alaa Salah Jumaah, Hawraa Sahib Al-Haddad, Akeel Abed Yasseen.

**Project administration:** Alaa Salah Jumaah, Hawraa Sahib Al-Haddad, Katherine Ann McAllister.

**Resources:** Alaa Salah Jumaah, Hawraa Sahib Al-Haddad, Katherine Ann McAllister.

**Software:** Alaa Salah Jumaah, Hawraa Sahib Al-Haddad.

**Supervision:** Alaa Salah Jumaah, Hawraa Sahib Al-Haddad, Katherine Ann McAllister, Akeel Abed Yasseen.

**Validation:** Hawraa Sahib Al-Haddad.

**Visualization:** Alaa Salah Jumaah, Hawraa Sahib Al-Haddad, Katherine Ann McAllister.

**Writing – original draft:** Alaa Salah Jumaah, Katherine Ann McAllister, Akeel Abed Yasseen.

**Writing – review & editing:** Katherine Ann McAllister, Akeel Abed Yasseen.

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
