## [Decision Letter · Decision Letter 0]

10 Nov 2021

PONE-D-21-17137The clinicopathology and survival characteristics of patients with POLE proofreading mutations in endometrial carcinoma: A systematic review and meta-analysis.PLOS ONE

Dear Dr. McAllister,

Thank you for submitting your manuscript to PLOS ONE. After careful consideration, we feel that it has merit but does not fully meet PLOS ONE’s publication criteria as it currently stands. Therefore, we invite you to submit a revised version of the manuscript that addresses the points raised during the review process.

We look forward to receiving your revised manuscript.

Kind regards,

Manish S. Patankar, Ph.D.

Academic Editor

PLOS ONE

Additional Editor Comments (if provided):

Two reviewers have carefully reviewed your manuscript. Please address the comments from the reviewers and submit your manuscript after making the modifications.

Journal Requirements:

Reviewers' comments:

Reviewer's Responses to Questions

**Comments to the Author**

1. Is the manuscript technically sound, and do the data support the conclusions?

Reviewer #1: Partly

Reviewer #2: Yes

2. Has the statistical analysis been performed appropriately and rigorously? 

Reviewer #1: No

Reviewer #2: Yes

3. Have the authors made all data underlying the findings in their manuscript fully available?

Reviewer #1: Yes

Reviewer #2: Yes

4. Is the manuscript presented in an intelligible fashion and written in standard English?

Reviewer #1: Yes

Reviewer #2: Yes

5. Review Comments to the Author

Reviewer #1: General comments: The authors present a systematic review and meta-analysis of the prevalence of POLE mutations in endometrial cancer, survival outcomes, and clinical/pathologic associations. This is an expansion upon a prior meta-analysis by the same authors, and though multiple meta-analyses on the same or similar topic have been performed, this does appear to have the highest number of patients with POLE mutated endometrial cancer in a meta-analysis to date.

- Previously published meta-analyses include those mentioned by the authors as well as the recently published McAlpine JN, Chiu DS, Nout RA, et al. Evaluation of treatment effects in patients with endometrial cancer and POLE mutations: An individual patient data meta-analysis, published in Cancer earlier this year.

Reviewer comments:

Major:

1. Starting in line 13 and throughout the paper, there are multiple comments regarding “improved survival” or “improved overall survival”; however, this 95% confidence interval crosses 1.

a. Also in line 313, 327.

2. Starting in line 18 (“invasion of the myometrium is curtailed,” citing only the significant p-value) and throughout the paper, the results regarding myometrial invasion seem to be overstated. While the OR for myometrial invasion <50% is 1.765 (95% CI 1.28-2.44, p=0.001), the OR for myometrial invasion >50% is 0.83 (95% CI 0.56-1.22, p=0.34), with a 95% CI crossing 1 and a p-value indicating a lack of significance, indicated that there is not (at least significantly) lower odds of deep (>50%) myometrial invasion with POLE mutations.

a. Another example of this is lines 243-245.

b. Also in line 328.

c. Also in line 360: “remain localized in the endometrium”

3. The association of POLE-mutant EC with endometrioid histology is reported as “significant”; however, the 95% CI crosses 1 and the p value is >0.05 (0.073). What are you using as your cut off for a significant p-value?

4. Lines 53-62: Citation 7 is cited when referring to your own prior meta-analysis, but also in sentences referring to other meta-analyses or results (at least some of which I can not find in your prior paper). Please cite the appropriate primary literature. Similarly, in lines 61-62, citation 7 is cited regarding depth of invasion; however, if this is referring to your prior meta-analysis, the 95% CI crosses 1.

5. Line 222 and forward: ORs should not be expressed as a “%.”

Minor:

1. Line 14-17: The way the data regarding stage and grade reads throughout the paper is confusing and needs to be rephrased. E.g. “Compared to wild type, POLE mutant tumors were significantly more likely to present at an early stage (stage I-II, OR=2.96, p<0.001) and high grade (grade 3, OR=1.72, p=0.003).”

a. Line 235-236: Another example. E.g. “POLE mutant EC are significantly more likely to present at an early stage (stage I-II) and high grade (grade 3) compared to wild type POLE EC.”

b. Lines 304-305 as well.

2. Line 51-52: There are papers regarding this subject. Please cite.

3. Line 71: Move “(2)” before “confirm.”

4. Would recommend having the paper reviewed for both grammar and spelling errors. There are many sentences that do not make sense as well as run-on and incomplete sentences.

5. Line 165: In multiple locations throughout the paper, where I believe you are intending to write “PFS” for progression free survival, there are typos such as “PFR” and “PFF.” Please correct.

6. Throughout: Consistently include a space between “95%” and “CI.”

7. Line 195, Table 3: Please clarify in the table legend that for the first line (overall POLE mutation), the denominator is all patients with EC; whereas for subsequent lines, the denominator is limited to patients with POLE mutated EC.

a. Also note typo/misspelling of “endometrioid” in the table.

8. Line 200: Adding “the proportion of” between “A subgroup analysis of” and “POLE mutant EC” will clarify this sentence.

9. Consider limiting decimal places to 2 as 3 can be bulky to read.

10. Lines 241-245: Rephrase for readability – and include p-values. E.g. “The odds ratio of MI in POLE mutant EC to MI in wild type POLE EC is 1.765 (95% CI: 1.28-2.44, p=0.001) for MI<50%, but 0.83 (95% CI: 0.56-1.22, p=0.34) for MI>50%.”

11. Line 248-249: “On the other hand” does not make sense here.

12. Lines 256 forward: The term “negatively involved lymph nodes” does not make sense to me. I would change this to negative lymph nodes and positive lymph nodes.

a. “The pooled proportion of positive lymph nodes in POLE mutant EC is 1.28% (95% CI: 0.24-3.84), while that for negative lymph notdes is 74.33% (95% CI: 61.79-85.11).

13. Line 258 and elsewhere: “odd ratio” should be changed to “odds ratio.”

14. Line 264: Insert “EC” between “type” and “in.”

15. Line 268: I would recommend deleting the word “great.”

16. Lines 278-295: This background information seems more appropriate for the introduction than the discussion.

17. Discussion: I would recommend decreasing the use of exact numbers as these have been presented in the results section, with a focus on a broader summary of the results and discussion of the implications. Would also not refer to tables in the discussion.

18. Line 309: Please clarify whether you are referring to your own previous results.

19. Line 316: What is the “main series of EC studies” referring to?

20. Lines 325-328: The flow from one sentence to the other makes it sound as though you have verified this reasoning/mechanism behind differences in survival outcomes.

21. Line 328-330: May want to discuss the published study(ies) regarding analysis of differences in treatment groups within the POLE mutant ECs.

22. Line 360: Recommend specifying lymph node metastases.

Reviewer #2: This is a well written meta-analysis that follows PRISMA guidelines.

The study aimed to: (1) resolve the reported clinicopathology variations of POLE-mutant endometrial carcinoma and confirm (2) the prognostic benefit of the POLE (exonuclease domain mutant) subtype using survival analysis of the expanded cohort of studies.

Specific suggestions:

Remove the word "fortunately" from line 88

Line 189- define the countries included, rather than "western countries"

ESMO risk stratification is not universally used and should be definted

It would be interesting to also report on tumor size, as that has always been linked to prognosis

6. PLOS authors have the option to publish the peer review history of their article (what does this mean?). If published, this will include your full peer review and any attached files.

Reviewer #1: No

Reviewer #2: No

---

## [Author Response · Author response to Decision Letter 0]

14 Dec 2021

RESPONSE TO REVIEWERS

Reviewer 1: 

The authors would like to thank reviewer one for constructive feedback of the manuscript.

Major:

1. Starting in line 13 and throughout the paper, there are multiple comments regarding “improved survival” or “improved overall survival”; however, this 95% confidence interval crosses 1.

a. Also in line 313, 327.

Responses: The authors have noted the reviewer’s comment regarding incorrect terminology with ‘improved overall survival’ when the 95% confidence intervals crosses 1. We have checked over the manuscript and amended this comment where applicable. The authors take on board that the results regarding myometrial invasion could be misinterpreted and have revised our conclusion according to the less advanced disease setting.

a. Lines 11-14: A total of 11 cohort studies of 5508 EC patients (442 EC POLE EDM) were included. POLE mutant EC was associated with improved survival: the pooled HR for overall, disease specific, and progression-free survival was 0.772 (95% CI: 0.574 to 1.039), 0.408 (95% CI: 0.306 to 0.543), and 0.231 (95% CI: 0.117 to 0.456) respectively. 

Lines 35-36 Revision: Patients with POLE mutant EC were associated with improved disease specific survival (HR=0.408, 95% CI: 0.306 to 0.543) and progression-free survival (HR=0.231, 95% CI: 0.117 to 0.456).

b. Line 169-171. OS was investigated in 442 cases of POLE mutant EC using data obtained from 11 studies. The OS associated with POLE mutant EC patients was favorable with a pooled HR of 0.772 (95%CI: 169 0.574 to 1.039) (Table 2). 

Paragraph Revisions Lines 217-229

OS was assessed using data extracted from ten studies, eight for DSS, and six for PFS, in patients with POLE mutant EC. The meta-analyses of hazard ratios for survival outcomes OS, DSS, and PFS are shown in Fig. 2A-C). 

Patients with POLE mutant tumors had associated improved survival, according to the parameters DSS (HR=0.408, 95% CI:0.306 to 0.543) and PFS (HR= 0.231, 95% CI: 0.117 to 0.456) (Table 2). However the OS outcome in POLE mutant EC patients was unclear (HR=0.772, 95% CI:0.574 to 1.039). While the HR was favorable for OS, the confidence interval had a wide range and crosses one.

c. Line 313. The meta-analysis determined favorable OS (HR: 0.772), DSS (HR: 0.408), and PFF (0.231) in the patients with mutant POLE EC. The findings of improved PFS and DSS in POLE mutant EC, confirms the results of prior studies [8, 10-16, 28-35]. 

Revision lines 389-395:

This meta-analysis also determined favorable PFS and DSS in POLE EDM tumors, confirming the results of many prior studies (8, 10-12, 15, 24, 25, 27, 28, 39). While the HR was also favorable for OS in the current investigation (HR=0.772) and consistent with previous studies including prior meta-analyses (20, 21) - the confidence interval had a wide range and crosses one (95% CI:0.574 to 1.039). Therefore the influence of POLE EC mutations on patient OS is still uncertain, and larger sample sizes are required to determine effects.

d. Line 327. Lines lines 323-327 PLOS PDF. This is clearly evidenced in the updated meta-analysis with the improved survival outcomes and majority clinicopathologic findings of: low FIGO stage and ESMO risk groups, lack of lymph node involvement, MI and LVI). 

Revision in lines 366-368:

This is clearly evidenced in the updated meta-analysis with the improved survival outcomes (DSS and PFS) and clinicopathologic findings (low FIGO stage and ESMO risk groups, lack of positive lymph node involvement, with no clear association with LVSI or deep MI).

2- Starting in line 18 (“invasion of the myometrium is curtailed,” citing only the significant p-value) and throughout the paper, the results regarding myometrial invasion seem to be overstated. While the OR for myometrial invasion <50% is 1.765 (95% CI 1.28-2.44, p=0.001), the OR for myometrial invasion >50% is 0.83 (95% CI 0.56-1.22, p=0.34), with a 95% CI crossing 1 and a p-value indicating a lack of significance, indicated that there is not (at least significantly) lower odds of deep (>50%) myometrial invasion with POLE mutations. 

Revisions become:

Lines 41-42. The tumors are significantly associated with invasion less than half (<50%) of the myometrium (OR=1.765, p=0.001), but not deeply invasive EC (MI>50%, OR=0.83, p=0.34). 

a. Another example of this is lines 243-245, PLOS PDF . “Our study showed that POLE mutation significantly curtails MI (p=0.001) to within <50% relative to wild type, which is an important prognostic parameter (Table 4)”.

Revisions become (lines 298-300):

POLE mutations are only significantly associated with invasion less than one half (<50%) of the myometrium (p=0.001) relative to wild type tumors (Table 4). However, this tendency is lost during advanced disease. There is no clear association of POLE mutations and deep MI >50% (OR=0.83, p=0.34).

b. Also in line 328, line 323-325 PLOS PDF. This is clearly evidenced in the updated meta- analysis with the improved survival outcomes and majority clinicopathologic findings of: low FIGO stage and ESMO risk groups, lack of lymph node involvement, MI and LVI). 

Revisions, lines 366-368:

This is clearly evidenced in the updated meta-analysis with the improved survival outcomes (DSS and PFS) and clinicopathologic findings (low FIGO stage and ESMO risk groups, lack of positive lymph node involvement, with no clear association with LVSI or deep MI). 

c. Also in line 360: “remain localized in the endometrium”. (The future treatment and well-being of POLE-mutant patients holds great promise. Our meta- analysis has determined that POLE-mutant tumors present at lower FIGO stage, are associated with the low ESMO risk group, and remain localized in the endometrium with reduced likelihood of metastasis).

Revisions become (discussion first main paragraph, lines 335 onwards):

Despite the alarming presentation with FIGO grade III tumors, patients with POLE mutant EC have improved PFS, DSS, and are mainly classified in the lowest ESMO risk group. Endometrial tumors with the POLE EDM subtype tumors are typically low FIGO stage. The clinicopathology meta-analysis findings also suggest that POLE mutant tumors remain localized at the endometrium, without deep progression into the muscular myometrial layer, with reduced likelihood of metastasis since the lymph nodes are mostly negative. In endometrial carcinoma, MI is a well-known predictor of recurrence, and an important factor in the decision-making process for adjuvant treatment (35). Patients with more than 50% MI on gross visual intraoperative estimation are at high risk for extrauterine metastases, including pelvic and para-aortic lymph node metastases (36). The current meta-analyses shows that POLE mutated EC tumor invasion is significantly curtailed to within 50% of the myometrium, there is no clear association with POLE mutations with invasion beyond 50% of the myometrium, and likelihood of positive lymph node metastases is reduced when compared to wild type POLE tumors (OR=0.202, p=0.001).

 3. The association of POLE-mutant EC with endometrioid histology is reported as “significant”; however, the 95% CI crosses 1 and the p value is >0.05 (0.073). What are you using as your cut off for a significant p-value?

Response The cut off for a significant p-value is >0.05. Revisions, (lines 319 onwards paragraph): The pooled proportion of endometrioid type in POLE mutant EC is 84.480% (95%CI: 77.237 to 90.548) while that for non-endometrioid type is 12.437 % (95% CI: .473 to 18.447). The pooled odd ratio of endometrioid histology in POLE mutant EC compared to wild type tumors is 1.386 (p=0.073) and that for the non-endometrioid variant is 0.582 (p=0.007). The findings suggest that POLE mutant EC have mainly endometrioid histology, but not significantly so when compared to wild type tumors (Refer to Tables 3, 4 and S8 Fig. A-D).

4. Lines 53-62: Citation 7 is cited when referring to your own prior meta-analysis, but also in sentences referring to other meta-analyses or results (at least some of which I can not find in your prior paper). Please cite the appropriate primary literature. Similarly, in lines 61-62, citation 7 is cited regarding depth of invasion; however, if this is referring to your prior meta-analysis, the 95% CI crosses 1.

Response: All references have been extensively checked and reinserted into the manuscript using ENDNOTE software to avoid such errors happening. The statement regarding depth of invasion has been removed. Citation 7 is now 21 (He et al 2020) in several places in the manuscript and is updated now.

5. Line 222 and forward: ORs should not be expressed as a “%.” Response: Done

Minor:

1. Line 14-17: The way the data regarding stage and grade reads throughout the paper is confusing and needs to be rephrased. E.g. “Compared to wild type, POLE mutant tumors were significantly more likely to present at an early stage (stage I-II, OR=2.96, p<0.001) and high grade (grade 3, OR=1.72, p=0.003).” 

Response: A fascinating characteristic of POLE mutant tumors is that they are high grade (loss of polymerase epsilon causes an ultramutated phenotype and greater likelihood of high grading); yet these tumors remain early-stage endometrial cancer and don’t metastasize. The confusion with presenting the results of stage and grade is gratefully noted by the authors - who have reworded stage and grade findings throughout. It is important that this finding is clearly communicated to readers.

a. Line 235-236: Another example. E.g. “POLE mutant EC are significantly more likely to present at an early stage (stage I-II) and high grade (grade 3) compared to wild type POLE EC.”

Response: Done

b. Lines 304-305 as well.

Response: Done

2. Line 51-52: There are papers regarding this subject. Please cite.

 Response: Done

3. Line 71: Move “(2)” before “confirm.”

4. Would recommend having the paper reviewed for both grammar and spelling errors. There are many sentences that do not make sense as well as run-on and incomplete sentences.

Response: Done

5. Line 165: In multiple locations throughout the paper, where I believe you are intending to write “PFS” for progression free survival, there are typos such as “PFR” and “PFF.” Please correct.

Response: A find and replace search was performed in word to replace PFF/PFR with PFS

6. Throughout: Consistently include a space between “95%” and “CI.”

Response: A find and replace updated the space.

7. Line 195, Table 3: Please clarify in the table legend that for the first line (overall POLE mutation), the denominator is all patients with EC; whereas for subsequent lines, the denominator is limited to patients with POLE mutated EC.

Table legend in “Clinicopathologic characteristics in EC”

Response: Done

Also note typo/misspelling of “endometrioid” in the table.

a. Response: Endometrioid Carcinoma of Endometrium is the most common histology subtype of endometrial cancer

b. 8. Line 200: Adding “the proportion of” between “A subgroup analysis of” and “POLE mutant EC” will clarify this sentence.

Response: Done

9. Consider limiting decimal places to 2 as 3 can be bulky to read.

10. Lines 241-245: Rephrase for readability – and include p-values. E.g. “The odds ratio of MI in POLE mutant EC to MI in wild type POLE EC is 1.765 (95% CI: 1.28-2.44, p=0.001) for MI<50%, but 0.83 (95% CI: 0.56-1.22, p=0.34) for MI>50%.”

POLE mutations are only significantly associated with MI (p=0.001) within <50% relative to wild type tumors (Table 4). However this tendency is lost when the disease becomes more advanced (MI >50%, p=0.34).

11. Line 248-249: “On the other hand” does not make sense here.

Done

12. Lines 256 forward: The term “negatively involved lymph nodes” does not make sense to me. I would change this to negative lymph nodes and positive lymph nodes.

a. “The pooled proportion of positive lymph nodes in POLE mutant EC is 1.28% (95% CI: 0.24-3.84), while that for negative lymph notdes is 74.33% (95% CI: 61.79-85.11).

Done, thank you for this suggestion

13. Line 258 and elsewhere: “odd ratio” should be changed to “odds ratio.”

Done with find and replace throughout manuscript.

14. Line 264: Insert “EC” between “type” and “in.”

Done

15. Line 268: I would recommend deleting the word “great.”

Done

16. Lines 278-295: This background information seems more appropriate for the introduction than the discussion.

Noted. The discussion material has been replaced in the introduction, first paragraphs.

17. Discussion: I would recommend decreasing the use of exact numbers as these have been presented in the results section, with a focus on a broader summary of the results and discussion of the implications. Would also not refer to tables in the discussion.

Responses: the authors agree with this and have completely revised the discussion to meet the required standards

18. Line 309: Please clarify whether you are referring to your own previous results.

Done (paragraph 377 onwards): The result of the current meta-analysis were also compared to previous studies, including our prior meta-analysis (20) and that of He and colleagues (21).

19. Line 316: What is the “main series of EC studies” referring to?

Responses: This sentence has been omitted, but did refer to the TCGA molecular classification of EC and later studies onwards

20. Lines 325-328: The flow from one sentence to the other makes it sound as though you have verified this reasoning/mechanism behind differences in survival outcomes.

The discussion is based on the outcomes of the current and previously published studies.

21. Line 328-330: May want to discuss the published study(ies) regarding analysis of differences in treatment groups within the POLE mutant ECs. This is a great suggestion, but we did not have enough space to discuss this in the current paper.

22. Line 360: Recommend specifying lymph node metastases

Done in both discussion and also abstract (line 25): The tumors are characteristically high grade, early stage, and remain localized in the endometrium with reduced likelihood of lymph node metastasis for improved survival prospects and the lowest risk classification.

Reviewer 2 comments:

The authors would like to thank reviewer two for constructive feedback of the manuscript.

1. Remove the word "fortunately" from line 88. Done

2. Line 189- define the countries included, rather than "western countries"

The line has been expanded to: All the studies were performed in Canada, the USA, or European countries. 

3. ESMO risk stratification is not universally used and should be definted.

Answer: Done

The material and methods has outlined the ESMO risk stratification in lines 150-159:

The ESMO guidelines categorizes the risk of recurrence into (1) low, (2) intermediate, and (3) high-risk groups by tumor stage, grade, and histology subtype (30). For example, about 75% of patients present with stage I disease and can be subdivided into three risk categories with regard to disease

relapse and survival:

(1) low risk: stage Ia/Ib, grade 1 or 2, endometrioid histology

(2) intermediate risk: stage Ic, grade 1 or 2, endometrioid histology; stage Ia/Ib, grade 3, endometrioid histology

(3) high risk: stage Ic, grade 3, endometrioid histology; stage Ia or Ib or Ic, serous, clear cell, small cell or undifferentiated histology

Furthermore the discussion has added content on treatment recommendations of the three ESMO risk groups (lines 368-373)

4. It would be interesting to also report on tumor size, as that has always been linked to prognosis

Response: Agreed, this would be interesting to report. Tumor size is linked to prognosis in many patients; and as part of the TNM staging system such as for breast carcinoma. However, there were insufficient data regarding tumor size in the endometrial carcinoma studies which are incorporated in this meta-analysis. Therefore, we could not perform meta-analysis in respect to tumor size. However, we have reported on tumor stage which will be affected by the size of the tumor.

---

## [Decision Letter · Decision Letter 1]

24 Jan 2022

The clinicopathology and survival characteristics of patients with POLE proofreading mutations in endometrial carcinoma: A systematic review and meta-analysis.

PONE-D-21-17137R1

Dear Dr. McAllister

We’re pleased to inform you that your manuscript has been judged scientifically suitable for publication and will be formally accepted for publication once it meets all outstanding technical requirements.

Kind regards,

Manish S. Patankar, Ph.D.

Academic Editor

PLOS ONE

Additional Editor Comments (optional):

Reviewers' comments:

Reviewer's Responses to Questions

**Comments to the Author**

1. If the authors have adequately addressed your comments raised in a previous round of review and you feel that this manuscript is now acceptable for publication, you may indicate that here to bypass the “Comments to the Author” section, enter your conflict of interest statement in the “Confidential to Editor” section, and submit your "Accept" recommendation.

Reviewer #1: All comments have been addressed

Reviewer #2: All comments have been addressed

2. Is the manuscript technically sound, and do the data support the conclusions?

Reviewer #1: Yes

Reviewer #2: Yes

3. Has the statistical analysis been performed appropriately and rigorously? 

Reviewer #1: Yes

Reviewer #2: Yes

4. Have the authors made all data underlying the findings in their manuscript fully available?

Reviewer #1: Yes

Reviewer #2: Yes

5. Is the manuscript presented in an intelligible fashion and written in standard English?

Reviewer #1: Yes

Reviewer #2: Yes

6. Review Comments to the Author

Reviewer #1: Congratulation for completing this important manuscript. All my comments have been addressed and manuscript appear much improved and ready for publication.

Reviewer #2: Thank you for incorporating the suggested feedback. This article is fit for publication. Well done.

7. PLOS authors have the option to publish the peer review history of their article (what does this mean?). If published, this will include your full peer review and any attached files.

Reviewer #1: No

Reviewer #2: No

---

## [Editor Report · Acceptance letter]

26 Jan 2022

PONE-D-21-17137R1 

The clinicopathology and survival characteristics of patients with POLE proofreading mutations in endometrial carcinoma: A systematic review and meta-analysis. 

Dear Dr. McAllister:

I'm pleased to inform you that your manuscript has been deemed suitable for publication in PLOS ONE. Congratulations! Your manuscript is now with our production department. 

Kind regards, 

on behalf of

Dr. Manish S. Patankar 

Academic Editor

PLOS ONE